# Transition Path Sampling with Boltzmann Generator-based MCMC Moves

## Abstract

Sampling all possible transition paths between two 3D states of a molecular system has various applications ranging from catalyst design to drug discovery. Current approaches to sample transition paths use Markov chain Monte Carlo and rely on time-intensive molecular dynamics simulations to find new paths. Our approach operates in the latent space of a normalizing flow that maps from the molecule's Boltzmann distribution to a Gaussian, where we propose new paths without requiring molecular simulations. Using alanine dipeptide, we explore Metropolis-Hastings acceptance criteria in the latent space for exact sampling and investigate different latent proposal mechanisms.

## 1 Introduction

Sampling the trajectories in which a molecular system changes from one 3D configuration to another—a task known as transition path sampling (TPS)—has many applications, such as designing catalysts [Crehuet and Field, 2007], materials [Selli et al., 2016], or drug discovery [Kirmizialtin et al., 2012, 2015]. In fact, the transition path ensemble is the ideal description of a chemical reaction's mechanism. We explore how this problem can be solved using a Boltzmann generator (a normalizing flow trained to sample a molecule's Boltzmann distribution) [Noé et al., 2019] and its latent space to obtain or approximate the ensemble.

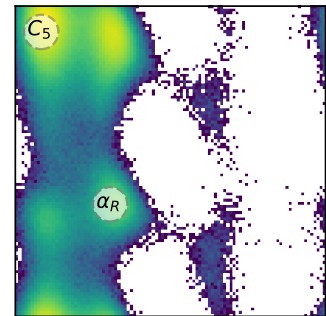

Figure 1: Distribution of alanine dipeptide's 3D configurations visualized via a histogram of its main dihedral angles $\phi, \psi$. Two metastable states are highlighted, between which we aim to sample the ensemble of all possible transition paths.

In the TPS problem, we are given a single molecular system and two 3D conformations of interest for it: states A and B, as seen in Figure 1. These could be the structure of reactants before a reaction and the structure of the product molecule after the reaction. With this, we aim to sample the transition paths between them with the likelihood at which they occur. To describe a transition path, we use a sequence of time-equidistant 3D atom configurations (i.e., frames) that starts in state A and ends in state B.

Existing approaches for this problem [Dellago et al., 1998b,a, Bolhuis et al., 2002] use Markov chain Monte Carlo (MCMC) sampling to iteratively sample a new path given the current one. New paths are commonly proposed using shooting moves that require molecular dynamics simulation. Given a path, the proposal is generated by first randomly selecting a frame of the path and sampling a random velocity from a Gaussian. The selected frame with the new velocity is then simulated forward and backward in time. If the backward simulation reaches state A and the forward simulation ends in state B, this trajectory constitutes a new non-zero probability transition path, which is accepted or

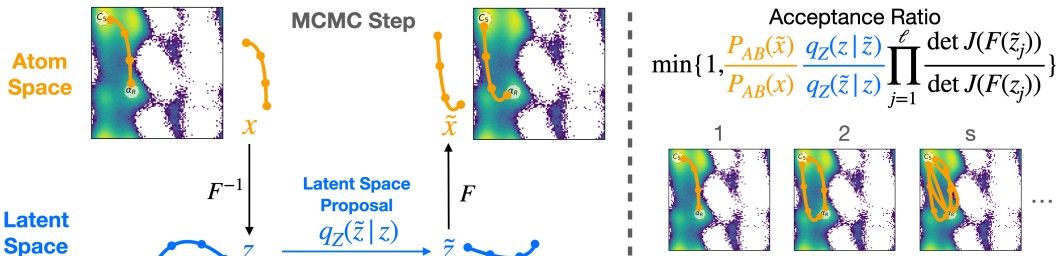

Figure 2: **MCMC proposals for latent space transition paths.** We move a transition path $x$ into latent space using a Boltzmann generator $F(\cdot)$. With this path $z$ and our latent space proposal kernel $q_Z(\tilde{z}|z)$ we propose $\tilde{z}$ and bring it back to configuration/atom space to obtain the transition path proposal $x$. The likelihood of all steps can be computed, and we use them in a Metropolis-Hastings acceptance criterion to sample the transition path ensemble with MCMC.

rejected based on its probability and a Metropolis-Hastings [Metropolis et al., 1953, Hastings, 1970] acceptance criterion. All paths that do not transition between A and B will be rejected. Repeating this is guaranteed to eventually produce the exact transition path ensemble, but convergence is slow since many proposals will not fulfill the constraints, paths are correlated, and finding transitions requires expensive simulation.

In this work, we explore how the TPS problem can be addressed when having access to a trained Boltzmann generator, which solves the easier problem of sampling the molecular systems distribution of 3D conformers. Given this, we generate MCMC proposals by first moving every frame in a path into our latent space. We modify each frame of this path by adding independent Gaussian noise such that the overall likelihood can easily be evaluated. Then, we use the Boltzmann generator to bring the whole path back to configuration space, compute the probability of the path, and use it to accept or reject the proposed path. This procedure is depicted in Figure 2.

Our contributions are investigating this novel method for transition path sampling and highlighting its challenges. To that end, we describe the difficulty of calculating likelihoods for paths that were not generated with molecular dynamics and the obstacles for calculating path probabilities in parallel. Additionally, we provide insights into what configuration space paths are produced from simple paths in the latent space of a Boltzmann generator.

## 2 Background and Related Work

**Boltzmann generators.** Given a molecule, the probability of each 3D configuration is proportional to the exponential of its negative energy, i.e., they follow a Boltzmann distribution. Noé et al. [2019] train a normalizing flow [Tabak and Vanden-Eijnden, 2010, Tabak and Turner, 2013] to sample a molecule's Boltzmann distribution, known as Boltzmann generator. While recent innovations [Midgley et al., 2023b,a] improved their training efficiency, training them for larger systems remains an open problem and a limitation of our Boltzmann generator-based approach.

**Deep learning for transition path sampling.** The TPS problem, with the goal to sample the whole transition path ensemble, is more challenging than finding a single low-energy transition path: A problem that also has been explored with deep learning (DL) approaches [Liu et al., 2022, Holdijk et al., 2023]. For the harder TPS problem, DL methods require MCMC with shooting moves as proposed by Dellago et al. [1998b,a]. For instance, Falkner et al. [2023] replace the shooting point selection with DL and sample them with a Boltzmann generator. Similarly, Jung et al. [2023] increase the acceptance rate of shooting moves by selecting the frames to shoot from with a learned function. These approaches still require sequential MD simulation. In this work, we explore a novel molecular dynamics-free MCMC paradigm using DL.

## 3 Method

We assume access to a Boltzmann generator for the molecule of interest and two of its states, A and B, between which we wish to sample the transition path ensemble. In the following, we lay out the overall MCMC framework over latent space paths (see § 3.1). This requires two components:

74 Calculating the path probability (see § 3.2), and a proposal kernel for a path in latent space for which
75 we lay out several options (see § 3.3).

## 3.1 MCMC Framework for Latent Paths

77 Let $\boldsymbol{x}$ be our current path with frames $\boldsymbol{x}_i \in \mathbb{R}^{n \times 3}$ and $i \in \{1, ..., l\}$, where $n$ is the number of atoms
78 of our molecule and $l$ the number of frames which we keep constant (the spacing of the frames
79 can change with changing path lengths). For our MCMC procedure, we further need a proposal
80 kernel $q(\tilde{\boldsymbol{x}} \mid \boldsymbol{x})$ that produces a new path proposal $\tilde{\boldsymbol{x}}$ from our current path $\boldsymbol{x}$[1]. If we can additionally
81 compute the probability of a path $p_{AB}$, we can sample the transition path ensemble with the MCMC
82 algorithm using Metropolis-Hastings acceptance criterion

$$\alpha = \min \left\{ 1, \frac{p_{AB}(\tilde{\boldsymbol{x}})}{p_{AB}(\boldsymbol{x})} \cdot \frac{q(\boldsymbol{x} \mid \tilde{\boldsymbol{x}})}{q(\tilde{\boldsymbol{x}} \mid \boldsymbol{x})} \right\}. \tag{1}$$

83 In our work, the proposal consists of first using a Boltzmann generator $F$ trained on the molecule
84 to move the path $\boldsymbol{x}$ into latent space to obtain the latent path $\boldsymbol{z} = \left\{ F^{-1}(\boldsymbol{x}_1), \ldots, F^{-1}(\boldsymbol{x}_l) \right\}$.
85 Subsequently, we make a proposal in latent space to obtain a new latent path $\tilde{\boldsymbol{z}}$ using the latent
86 proposal kernel $q_z(\tilde{\boldsymbol{z}} \mid \boldsymbol{z})$ which we design in § 3.3. Lastly, the latent path is projected back to
87 configuration space using the Boltzmann generator $\boldsymbol{x} = \{ F(\tilde{\boldsymbol{z}}_1), \ldots, F(\tilde{\boldsymbol{z}}_l) \}$.

88 The proposal kernel thus takes the form $q(\tilde{\boldsymbol{x}} \mid \boldsymbol{x}) = p(\boldsymbol{z}|\boldsymbol{x})q_z(\tilde{\boldsymbol{z}} \mid \boldsymbol{z})p(\tilde{\boldsymbol{x}}|\tilde{\boldsymbol{z}})$, where $p(\boldsymbol{z}|\boldsymbol{x})$ accounts
89 for the change of density when using our Boltzmann generator to move the path $\boldsymbol{x}$ into latent space
90 and $p(\tilde{\boldsymbol{x}}|\tilde{\boldsymbol{z}})$ arises from moving the new latent path back to configuration space. Since the Boltzmann
91 generator processes all the frames independently, the change of density factors can be written as
92 the product of the individual frames $p(\boldsymbol{z}|\boldsymbol{x}) = \prod_{i=1}^{l} p(\boldsymbol{z}_i|\boldsymbol{x}_i)$. With this in mind, the ratio of the
93 forward path proposal $q(\tilde{\boldsymbol{x}} \mid \boldsymbol{x})$ and the backward proposal $q(\boldsymbol{x} \mid \tilde{\boldsymbol{x}})$, as it is required in the acceptance
94 criterion in Equation 1, takes the form

$$\frac{q(\boldsymbol{x} \mid \tilde{\boldsymbol{x}})}{q(\tilde{\boldsymbol{x}} \mid \boldsymbol{x})} = \frac{q_Z(\boldsymbol{z} \mid \tilde{\boldsymbol{z}})}{q_Z(\tilde{\boldsymbol{z}} \mid \boldsymbol{z})} \cdot \prod_{i=1}^{l} \frac{p(\tilde{\boldsymbol{z}}_i|\tilde{\boldsymbol{x}}_i)p(\boldsymbol{x}_i|\boldsymbol{z}_i)}{p(\boldsymbol{z}_i|\boldsymbol{x}_i)p(\tilde{\boldsymbol{x}}_i|\tilde{\boldsymbol{z}}_i)}. \tag{2}$$

95 Each term in the product can be simplified as follows, where we write $\boldsymbol{x}, \boldsymbol{z}$ for an individual frame
96 $\boldsymbol{x}_i, \boldsymbol{z}_i$ and use the change of variables formula $p(\boldsymbol{x}) = p(\boldsymbol{z}) \cdot (\det J(F(\boldsymbol{z})))^{-1}$ in the third equality

$$\frac{p(\tilde{\boldsymbol{z}}|\tilde{\boldsymbol{x}})p(\boldsymbol{x}|\boldsymbol{z})}{p(\boldsymbol{z}|\boldsymbol{x})p(\tilde{\boldsymbol{x}}|\tilde{\boldsymbol{z}})} = \frac{\frac{p(\tilde{\boldsymbol{x}},\tilde{\boldsymbol{z}})}{p(\tilde{\boldsymbol{x}})}\frac{p(\boldsymbol{x},\boldsymbol{z})}{p(\boldsymbol{z})}}{\frac{p(\boldsymbol{x},\boldsymbol{z})}{p(\boldsymbol{x})}\frac{p(\tilde{\boldsymbol{x}},\tilde{\boldsymbol{z}})}{p(\tilde{\boldsymbol{z}})}} = \frac{p(\boldsymbol{x})p(\tilde{\boldsymbol{z}})}{p(\tilde{\boldsymbol{x}})p(\boldsymbol{z})} = \frac{p(\boldsymbol{z})(\det J(F(\boldsymbol{z})))^{-1}p(\tilde{\boldsymbol{z}})}{p(\tilde{\boldsymbol{z}})(\det J(F(\tilde{\boldsymbol{z}})))^{-1}p(\boldsymbol{z})} = \frac{\det J(F(\tilde{\boldsymbol{z}}))}{\det J(F(\boldsymbol{z}))}. \tag{3}$$

97 Thus, the ratio of proposals we need to calculate is

$$\frac{q(\boldsymbol{x} \mid \tilde{\boldsymbol{x}})}{q(\tilde{\boldsymbol{x}} \mid \boldsymbol{x})} = \frac{q_Z(\boldsymbol{z} \mid \tilde{\boldsymbol{z}})}{q_Z(\tilde{\boldsymbol{z}} \mid \boldsymbol{z})} \cdot \prod_{j=1}^{l} \frac{\det J(F(\tilde{\boldsymbol{z}}_j))}{\det J(F(\boldsymbol{z}_j^{(i-1)}))}, \tag{4}$$

98 which we can readily use to calculate the acceptance ratio for the MCMC algorithm as laid out in
99 Algorithm 1. The remaining challenges are the ability to compute the path probability $p_{AB}$ and a
100 concrete latent space proposal kernel $q_Z(\tilde{\boldsymbol{z}} \mid \boldsymbol{z})$, which we will tackle next.

## 3.2 Calculating the Path Probability

102 A path's probability is defined with respect to a molecular dynamics model. Here, we assume
103 Langevin dynamics[2] under which the transition from frame $\boldsymbol{x}_i$ to the next frame $\boldsymbol{x}_{i+1}$ can be
104 calculated as

$$\mathbf{x}_{i+1} = \mathbf{x}_i + \Delta t \mathbf{v}_{i+1}$$
$$\mathbf{v}_{i+1} = \alpha \mathbf{v}_i + (1 - \alpha)\nabla U(\boldsymbol{x}_i) + \sqrt{k_B T(1 - \alpha^2)}\boldsymbol{W}, \tag{5}$$

---

[1]An initial path can be obtained from, for example, a high-temperature MD simulation or by linearly interpolating in the Boltzmann generator's latent space.

[2]For the sake of brevity, we omit the constant atom masses and the friction coefficient.

given the velocity $\boldsymbol{v}_i$, and the molecule's energy function $U : \mathbb{R}^{d \times 3} \mapsto \mathbb{R}$. $\alpha = \exp(-\Delta t)$ for a time step size $\Delta t$[3] and $\boldsymbol{W} \sim \mathcal{N}(0, \mathbb{1})$ corresponds to random motion that is scaled proportional to the Boltzmann constant $k_B$ and temperature $T$. Notice that in Langevin dynamics, the only randomness when obtaining $\boldsymbol{x}_{i+1}$ from $\boldsymbol{x}_i$ given the velocity $\boldsymbol{v}_i$ stems from the Gaussian variable $\sqrt{k_B T (1 - \alpha^2)}\boldsymbol{W}$. Thus, the probability density $p(\boldsymbol{x}_{i+1}, \boldsymbol{v}_{i+1} | \boldsymbol{x}_i, \boldsymbol{v}_i)$ of moving from $\boldsymbol{x}_i$ to $\boldsymbol{x}_{i+1}$ is that of a Gaussian with mean $\mu = \boldsymbol{x}_i + \Delta t(\alpha \boldsymbol{v}_i + (1 - \alpha)\nabla U(\boldsymbol{x}_i))$ and standard deviation $\sigma = k_B T (1 - \alpha^2)$.

Given this probability $p(\boldsymbol{x}_{i+1}, \boldsymbol{v}_{i+1} | \boldsymbol{x}_i, \boldsymbol{v}_i)$ of moving between individual frames with the auxiliary velocity variable, the probability of a whole path in configuration space is

$$p_{AB}(\boldsymbol{x}) = p(\boldsymbol{x}_1) \cdot \prod_{i=1}^{l-1} p(\boldsymbol{x}_{i+1}, \boldsymbol{v}_{i+1} | \boldsymbol{x}_i, \boldsymbol{v}_i), \tag{6}$$

where $p(\boldsymbol{x}_i)$ follows the molecule's Boltzmann distribution, meaning that $p(\boldsymbol{x}_i) \propto exp(-U(\boldsymbol{x}_i)/k_B T)$ with an unknown proportionality constant. However, this constant is unnecessary since it will cancel out with the same constant of the reverse path density $p_{BA}$ in the acceptance ratio in Equation 1.

Thus, the last missing link to computing $p_{AB}(\boldsymbol{x})$ is the initial velocity $\boldsymbol{v}_1$. Since our path definition does not include an initial velocity (because we do not have a Boltzmann generator that operates over both velocities and positions), we opt to marginalize over all possible velocities and approximate the following expectation as our final path probability

$$p_{AB}(\boldsymbol{x}) = \mathbb{E}_{\boldsymbol{v}_1 \sim \mathcal{N}(\mathbf{0}, \mathrm{diag}(k_B T))}\left[ p(\boldsymbol{x}_1) \cdot \prod_{i=1}^{l-1} p(\boldsymbol{x}_{i+1}, \boldsymbol{v}_{i+1} | \boldsymbol{x}_i, \boldsymbol{v}_i) \right]. \tag{7}$$

All subsequent velocities $\{\boldsymbol{v}_i\}_{i \in \{2, \ldots, l\}}$ can then be inferred by solving the previous step, allowing us to compute $p(\boldsymbol{x}_{i+1}, \boldsymbol{v}_{i+1} | \boldsymbol{x}_i, \boldsymbol{v}_i)$ sequentially.

**Desirable properties.** In designing our MCMC procedure, we set out to avoid the time-consuming sequential molecular dynamics simulation. While the path probability can be computed easily for paths generated by MD [Jung et al., 2017], calculating the path probability $p_{AB}(\boldsymbol{x})$ still requires sequential computation in our approach. However, this amounts to sequentially performing $l$ vector additions, which is very cheap and can be done in parallel for all different initial velocities when approximating the expectation. The expensive, time-consuming computations stem from the evaluation of the energy function $U(x_i)$ for each frame. In our procedure, this can be done in parallel, while in molecular dynamics, it has to be performed sequentially.

### 3.3 Latent Space Path Proposal Kernel

As for the concrete latent space path proposal kernel $q_Z(\tilde{z} \mid z)$, we propose three different options: 1) Gaussian noise added to each frame. 2) A Gaussian Process (GP) with the current path as its mean. 3) A GP that is adaptively fit to the history of all sampled transition paths and only weakly depends on the current path. All these proposals are symmetric and will not contribute to our acceptance ratio with $q_Z(\tilde{z} \mid z)/q_Z(z \mid \tilde{z}) = 1$.

**Gaussian proposal.** From a latent path $z$, we propose a new path $\tilde{z} = \{z_1 + \epsilon_1, \ldots, z_l + \epsilon_l\}$ where $\epsilon_1, \ldots, \epsilon_l \sim \mathcal{N}(\mathbf{0}, \boldsymbol{\Sigma})$. While this independent noise for each frame makes it unlikely that all frames move coherently and produce high-probability paths, this operation can be performed efficiently and allows for fast exploration of the latent space.

**Conditional Gaussian process path proposals.** We employ a GP $f(t) \sim \mathcal{GP}(m(t), k(t, t'))$, where $f : \mathbb{R} \mapsto \mathbb{R}^{3n-6}$ maps the time $t \in [1, l]$ along the path to a Gaussian from which a frame at time $t$ is sampled[4]. We fit the GP mean $m(\cdot)$ and kernel function $k(\cdot, \cdot)$, which is not to be confused with the proposal kernel, to a set of $s$ latent paths $\{z^i\}_{i \in \{1, \ldots, s\}}$, where the index of each frame is used as the

---

[3]Similar to classical fixed length transition path sampling, the timestep size $\Delta t$ is not trivial to choose. We discuss this further in Appendix B.

[4]The latent space dimensionality is $\mathbb{R}^{3n-6}$ for $n$ atoms since the Boltzmann generator operates on internal coordinates that are invariant to the 6 degrees of freedom from rigid translations and rotations.

time $t$. In the following, we first detail the set of latent paths before explaining how the GP is used to propose a new path.

Our set of latent paths $\{z^i\}_{i\in\{1,\dots s\}}$ to fit the GP is either the history of all previously sampled paths or we obtain it via linear interpolation in latent space. Specifically, to obtain an interpolation, we sample a start $x_1$ and an end frame $x_l$ from states A and B, move them to latent space to generate $z_1, z_l = F^{-1}(x_1), F^{-1}(x_l)$, and produce the latent path as the linear interpolation $z_i = \frac{i}{l}z_1 + (1 - \frac{i}{l})z_l$ for $i \in \{1, \dots, l\}$. After moving it back to configuration space with the Boltzmann generator, this constitutes a coarse path. This produces a fixed proposal kernel, where the quality depends on the paths it was trained on.

When using the history of all previously sampled paths as $\{z^i\}_{i\in\{1,\dots s\}}$, the proposal kernel $\mathcal{GP}_s$ changes over the course of MCMC steps $s$, leading to an adaptive MCMC algorithm. For this to be correct, the proposal kernel has to converge and satisfy vanishing adaptation [Andrieu and Thoms, 2008] where, as the Markov chain progresses, the influence of its most recent states on the proposal kernel has to diminish. Intuitively, this is the case for our adaptive kernel since the influence of the most recent path on the fitted mean and covariance kernel vanishes as the size of the history (the Markov chain) increases.

We re-fit this adaptive GP proposal to the history of latent paths $\{z^i\}_{i\in\{1,\dots s\}}$ at each step $s$ when a new path has been accepted. To efficiently do so, we start optimization from the parameters of the previous GP proposal kernel that are optimal for $\{z^i\}_{i\in\{1,\dots s-1\}}$. The new optimization's convergence is typically fast since the minimum under the new set of latent paths at step $s$ is likely close to that at step $s - 1$, with the difference diminishing as the length of the Markov chain increases.

Given the fitted GP, a new latent path $\tilde{z}$ is proposed conditioned on the current one $z$ by sampling $\mathcal{GP}_s$ at times $t = 1, \dots, l$ (which correspond to the frame numbers of the paths) after setting the means of $\mathcal{GP}_s$ at those times to the frames of $z$, meaning that $m(t) = z_t$ for $t \in \{1, \dots l\}$. This amounts to sampling $\mathcal{GP}_s$ unconditionally at $t = 1, \dots, l$, subtracting the means $m(t)$, and adding the frames $z_t$ at each time.

**Unconditional Gaussian process path proposals.** Here, we use the adaptive Gaussian process $\mathcal{GP}_s$ and propose new paths $\tilde{z}$ unconditionally, meaning that each proposal is a sample of $\mathcal{GP}_s$ and the only influence of $z$ is through its presence in the set of paths $\{z^i\}_{i\in\{1,\dots s\}}$ that $\mathcal{GP}_s$ was fit on. This means that with a progressively increasing number of accepted paths, the influence of the current path will diminish. This would fit a Gaussian process that could be used to sample transition paths without any latent space, which is an interesting aspect on its own.

Further, since we will rely on the mean of the Gaussian process, we can also estimate it between the frames. This allows us to introduce more variance by evaluating the Gaussian process not at the fixed points $1, \dots, l$, but to uniformly draw $l$ sorted samples from $\mathcal{U}_{[0.5,l+0.5]}$. With this, the individual frames of the path can shift more easily towards and from each other.

# 4    Experiments

**Latent space analysis.** When moving configurations from the meta-stable states $C_5$ and $\alpha_R$ of alanine dipeptide (ALDP) into the latent space, we can linearly interpolate between them and map them back with the trained Boltzmann generator. For this, we train a Boltzmann generator by minimizing the forward KL-divergence loss (compare Appendix A.3). Figure 3 shows that linear paths in latent space produce non-linear paths in configuration space. While linearly interpolating atom positions of a molecule produces unrealistic paths, this naive latent space approach recovers two different modes of transitions between the meta-stable states.

**Ground truth ensemble.** We simulated 10 nanoseconds with a timestep of 1 femtosecond at 300K with the openMM MD engine [Eastman et al., 2017]. From this data, we can determine for each conformation whether it belongs to a meta-stable state, allowing us to find paths by looking for sequences that start in A and transition to B (or vice versa). This approach finds variable-length transition paths. We rely on the two-way shooting scheme implemented by OPS [Swenson et al., 2018a,b] with the same MD setup to sample a fixed-length transition path ensemble. Transitions that only rarely occur (Figure 4 bottom) are particularly difficult to produce with classical MD, already for the small molecule alanine dipeptide.

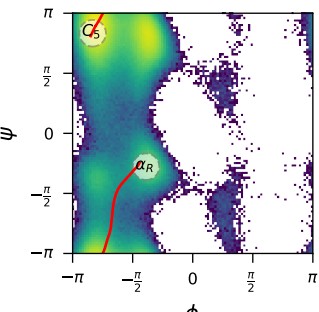
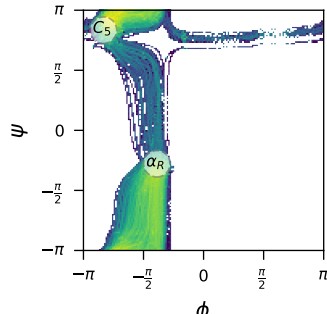

Figure 3: **Linear latent space interpolation.** *Left*: A histogram of the two main dihedral angels $\phi, \psi$ as they occur in the MD simulation. The meta-stable states $C_5$ and $\alpha_R$, and a linear interpolation in latent space (red line) are shown. *Right*: The resulting density of transition paths when linearly interpolating between those states in latent space.

**Results.** Figure 4 shows for all methods a histogram of the sampled transition paths between the states $C_5 \leftrightarrow \alpha_R$ and $\alpha_R \leftrightarrow C_7$, respectively. *Unconditional GP Uni* refers to the adaptive GP proposal with uniform timepoint sampling while *Unconditional GP* always samples the index of the frame as timepoint. *Conditional GP* uses the adaptive proposal.

The main finding is that due to the low acceptance rate of our MCMC steps, we are only able to produce a low amount of paths or a set of paths with low diversity. When increasing the variance, paths will be more diverse but are also less likely to be accepted. To overcome this, proposals that produce more physically likely paths are required.

Some proposal strategies, such as the Gaussian proposal, are computationally efficient, while fitting a high-dimensional Gaussian process is time-consuming. With an increasing number of paths, the proposals are more likely to be stuck in a local minimum. For the more computationally expensive proposals, this makes it challenging to produce the high number of transitions needed to overcome this threshold.

While training a fixed Gaussian process on simple paths in latent space is computationally favorable, the results do not indicate that it can capture the transition paths. Since the iterative Gaussian processes do not seem to fit the distribution either, our choice of kernel or formulation might be inappropriate. In general, we have seen in our experiments that the selection for a kernel of the Gaussian process (compare Appendix A.2) poses a difficult problem for this task because it must capture an adequate amount of noise without overfitting to the previous paths.

Overall, the results qualitatively show that the simplest proposal kernel, one that simply adds Gaussian noise in latent space, appears to be the most efficient and effective choice. Further, conditioning the Gaussian process on the current path appears to slightly increase the variance and leads to a more diverse set of paths.

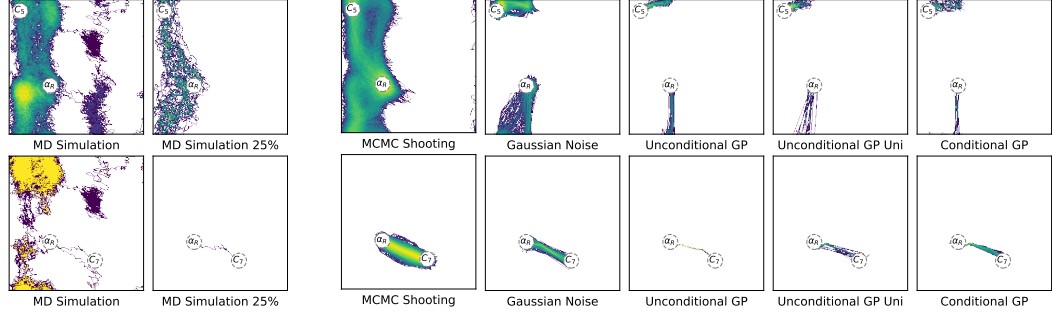

Figure 4: **Comparison of sampling methods**. Each row shows the transitions between two different metastable states. *Left*: "Ground truth" path ensemble from MD simulation of all paths (sub-left) and the 25% of paths with the highest probability (sub-right). *Right:* Shooting move MCMC ensemble and the ensembles of our different latent space proposal kernels. Note that it is unclear what a meaningful ground truth ensemble is.

## 5 Discussion and Conclusion

**Limitations** Our approach relies on a trained Boltzmann generator, of which high-quality ones for larger molecular systems do not exist yet. Furthermore, the latent space path proposal kernels we devise have too low acceptance rates to be useful. This limits them to a low-variance, slowing down mode-mixing. Better latent space proposals would be necessary. Lastly, an avenue toward a practical solution could be adaptively fine-tuning the Boltzmann generator to make simple paths in latent space correspond to physical paths that obey Langevin dynamics in configuration space.

**Conclusion** In this paper, we presented a novel way to propose transition paths in the latent space of a Boltzmann generator. Throughout this work, we have introduced multiple latent space path proposal kernels that perform (learned) operations. This enables a transition path sampling MCMC procedure without the need for molecular dynamics simulation. We believe that learned transition path sampling methods and, in general, simulation-free MCMC approaches are interesting research questions to explore and might lead to faster sampling methods.

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

## A Method Details

### A.1 Latent Path MCMC Algorithm

Our latent space path sampling approach builds on the Metropolis-Hastings method but relies on a modified acceptance criteria and an adapted proposal kernel.

---

**Algorithm 1:** Fixed-length latent space transition path sampling.

---

**Input:** Initial path $\boldsymbol{x}^{(0)}$ with $l$ frames, a trained Boltzmann generator consisting of the map $F$ and its inverse $F^{-1}$, the number of steps to run $N$, and a latent proposal kernel $K$ with proposal probability $q_Z\left(\cdot \mid \cdot\right)$.

**Output:** MCMC samples following target distribution $\left\{\boldsymbol{x}^{(1)}, \ldots, \boldsymbol{x}^{(N)}\right\}$.

1 Calculate latent space representation of initial path $\boldsymbol{z}^{(0)} = \left\{F^{-1}\left(\boldsymbol{x}_1^{(0)}\right), \ldots, F^{-1}\left(\boldsymbol{x}_l^{(0)}\right)\right\}$.

2 **for** $i \leftarrow 1 \ldots N$ **do**

3     **repeat**

4         Propose new path in latent space $\tilde{\boldsymbol{z}} = K\left(\boldsymbol{z}^{(i-1)}\right)$.

5         Compute the proposed path in configuration space $\tilde{\boldsymbol{x}} = \{F\left(\tilde{\boldsymbol{z}}_1\right), \ldots, F\left(\tilde{\boldsymbol{z}}_l\right)\}$.

6         Compute acceptance probability

$$\alpha = \min\left\{1, \frac{p_{AB}\left(\tilde{\boldsymbol{x}}\right)}{p_{AB}\left(\boldsymbol{x}^{(i-1)}\right)} \cdot \frac{q_Z\left(\boldsymbol{z}^{(i-1)} \mid \tilde{\boldsymbol{z}}\right)}{q_Z\left(\tilde{\boldsymbol{z}} \mid \boldsymbol{z}^{(i-1)}\right)} \cdot \prod_{j=1}^{l} \frac{\det J\left(F\left(\tilde{\boldsymbol{z}}_j\right)\right)}{\det J\left(F(\boldsymbol{z}_j^{(i-1)})\right)}\right\}.$$

7         Draw a uniformly distributed random number $u \sim \mathcal{U}_{[0,1]}$.

8     **until** *proposed path $\tilde{\boldsymbol{x}}$ is reactive* **and** $u \leq \alpha$;

9     Accept proposed path $\boldsymbol{z}^{(i)} = \tilde{\boldsymbol{z}}, \boldsymbol{x}^{(i)} = \tilde{\boldsymbol{x}}$.

10 **end**

---

### A.2 Gaussian Process Kernel

A Gaussian process fits the parameters of a kernel $k$. As for the concrete choice of kernel, we have decided to use an RBF-Kernel with an additional White kernel that can capture variance in the individual points. It can be formulated as

$$k(x, x') = c \cdot \exp\left(-\frac{\|x - x'\|_2^2}{2l^2}\right) + n \cdot \mathbb{1}_{x \neq x'}, \tag{8}$$

with learnable parameters $l, c, n$.

### A.3 Boltzmann Generator Training

We trained a Boltzmann generator $F$ on the molecule ALDP, consisting of multiple neural spline layers [Durkan et al., 2019] with a randomly masked coupling architecture between them. The coupling layers allow us to use arbitrarily complex neural networks, that do not have to be invertible while still allowing the overall function to be invertible [Dinh et al., 2017]. In this architecture, the neural network learns to predict 8 knots and the parameters of a quadratic rational spline function. Overall, we use 12 of these neural spline coupling layers each using a residual block with two layers with 256 hidden units. The performance of the trained Boltzmann generator is illustrated in Figure 5.

To train the normalizing flow, we use the samples from the long-running MD simulation of ALDP and maximize the likelihood of the frames in latent space is. The goal of the loss is that the samples in latent space are distributed according to the base distribution. This is achieved by minimizing the forward KL divergence

$$\mathcal{L}_{KL}\left(\boldsymbol{\theta}\right) \propto -\mathbb{E}_{x \sim X}\left[\log\left(p_u\left(F_{\boldsymbol{\theta}}^{-1}(x)\right)\right) - \log\left(\det J\left(F_{\boldsymbol{\theta}}^{-1}(x)\right)\right)\right]. \tag{9}$$

$J$ represents the Jacobian and $p_u$ is the distribution of our latent space. $F$ represents the invertible function of the Botlzmann generator that maps between the ground truth data distribution $X$ and is parameterized by $\boldsymbol{\theta}$.

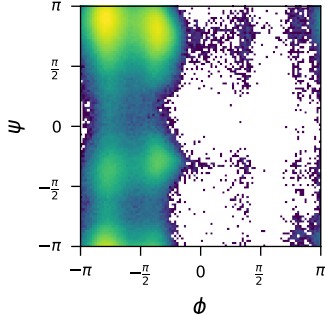

Figure 5: **Histogram of states sampled by Boltzmann generator.** This histogram shows the main dihedral angles of 1 million ALDP conformations sampled from the base distribution and then transported with the normalizing flow.

330  To represent the molecule, we rely on an internal coordinate representation for our flow, which
331  describes the molecule's state by the dihedral angles and bond lengths [Rezende et al., 2020] as this
332  has shown good performance [Noé et al., 2019, Midgley et al., 2023b]. Since some of these variables
333  are periodic, we use a mixture between a Gaussian and a uniform distribution as the base distribution.
334  This mixture is only used for training; at inference, we change this to a standard normal distributed
335  space by using the cumulative and inverse cumulative function to map uniform values from and to a
336  normal distribution.

337  **A.4  Further Latent Space Investigation**

338  To ensure that the learned latent space is meaningful and can separate between different meta-stable
339  states, we have reduced samples of the states $C_5$ and $\alpha_R$ to two dimensions, as seen in Figure 6.
340  Already a PCA, a non-linear dimensionality reduction, is capable the separating the states by a single
341  dimension. This motivates that a linear interpolation between configurations in latent space can
     produce feasible transition paths.

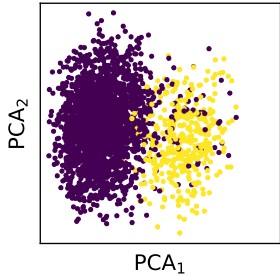 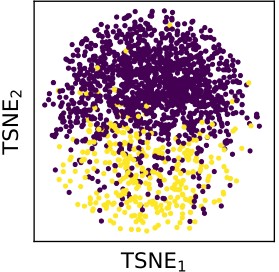

Figure 6: **Separability of meta-stable States in latent space.** Transforming molecule conformations following the states $C_5$ and $\alpha_R$ into the latent space, and plotting them in 2D with PCA and TSNE. The colors indicate the different conformations.

342

# B Determining the Timestep for TPS

Finding out the transition time between two states is necessary to be able to determine a suitable timestep $\Delta t$ and the number of frames. While this task can be challenging for large systems, this is not a task we set out to solve. To determine meaningful values, we have estimated the density of the transition times as they occur in a long-running MD simulation, as can be seen in Figure 7. With this, we have decided to sample transition paths with a duration of $1.6ps$. Similar studies can be performed to determine the transition times with high probability for transitions between $\alpha_R$ and $C_7$, where we decided to use a time of $320fs$.

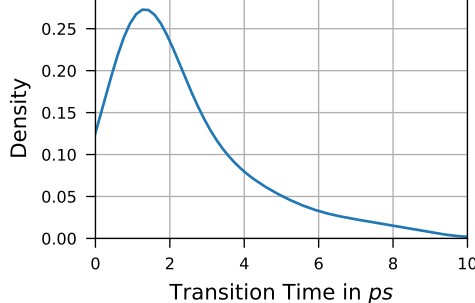

Figure 7: **Duration of ALDP transitions.** This is the approximated density that shows the duration of the transition between the states $C_5$ and $\alpha_R$ and their respective densities.