# OpenReview forum: "Transition Path Sampling with Boltzmann Generator-based MCMC Moves"
_NeurIPS.cc/2023/Workshop/AI4Science — NeurIPS2023-AI4Science Poster_

### Official Review · Reviewer_QJQW · 2023-10-15
**MCMC Transition Path Sampling with Learned Latent Space Proposals**

**Rating:** 7
**Confidence:** 5

**Review:**

This paper explores using a pre-trained Boltzmann generator's latent space to propose transition paths in MCMC sampling without molecular dynamics. It investigates Metropolis-Hastings acceptance criteria and different latent proposal methods like Gaussian noise and Gaussian processes. Experiments on alanine dipeptide highlight challenges around path probability calculation and low acceptance rates.

Quality: The methodology appears technically sound overall. The setup of the MCMC framework and derivations for path probabilities are clear. More analysis quantifying tradeoffs between proposals would further strengthen quality.

Clarity: The paper is well-written and easy to follow. The background gives sufficient context, and the methodology sections explain the approach clearly.

Originality: The idea of latent space MCMC for transition path sampling is interesting. While limitations remain, the work explores a creative simulation-free approach to this problem. It also provides unique insights into how simple latent paths map to complex configuration space paths.

Significance: The insights on learned latent space proposals are likely not generalizable, depending on how well the Boltzmann generator is trained. Also, as the authors state, it might not always be feasible to have access to a trained model.

---

### Official Review · Reviewer_XCMD · 2023-10-24
**Review of "Transition Path Sampling with Boltzmann Generator-based MCMC Moves**

**Rating:** 7
**Confidence:** 3

**Review:**

This work looks at a novel technique for transition path sampling (TPS) between two 3D states of a molecular system (alanine dipeptide). In particular, they propose new paths without requiring molecular simulations by operating in the latent space of a normalizing flow which defines a map between the molecule's Boltzmann distribution and a Gaussian distribution.

To describe a transition path, they use a sequence of time-equidistant 3D atom configurations between two states (A and B). Traditional approaches typically employ MCMC sampling to iteratively sample a new path given the current one which is then either accepted or rejected according to the Metropolis-Hastings (MH) criteria. Because of this, existing approaches are typically slow to converge and produce correlated paths, requiring significant computational effort and ultimately limiting their usefulness.

Assuming access to a Boltzmann generator for the molecule of interest (as well as the two states at the ends of the transition), they generate MCMC proposals by moving every frame in a path into their latent space. Each frame then has Gaussian noise added to it such that the overall likelihood can be easily evaluated. The Boltzmann generator is then used to return back to configuration space where the likelihood can be evaluated and either accepted or rejected according to the MH criteria.

An important property of the proposed approach is that it allows for the evaluation of the energy function, $U(x_{i})$ to be evaluated in parallel. This provides an important advantage over traditional molecular dynamics, where these evaluations must be carried out sequentially. They also go on to describe the challenges associated with this approach, and discuss the difficulty in evaluating likelihood ratios for the proposal paths.

They propose three different options for the latent space path proposal kernel $q_{Z}(\tilde{z} | z)$: (1.) Gaussian noise, (2.) Gaussian Process (GP) with current path as its mean, and (3.) A GP that is adaptively fit to the history of all sampled transition paths (weakly depending on the current path). Its worth noting that all of these proposals are symmetric and therefore will not contribute to the likelihood ratio in the MH accept / reject.

Ultimately, they find that due to the low acceptance rate of their MCMC steps, they are only able to produce a low amount of paths or a set of paths with low diversity. They discuss the cost / benefit analysis of their different proposal mechanisms, explaining clearly the limitations of each. Overall, their results suggest that the simplest proposal kernel (Gaussian noise in the latent space) is the best choice out of the three.

Overall, while lacking strong or decisive results, I still believe this is an interesting approach and worth exploring further. The main challenges associated with sequential MCMC (simulation cost, autocorrelations, inability to efficiently explore multi-modal targets) are pervasive, and the investigation of alternative approaches is a worthwhile endeavor.

---

### Meta-Review · Area_Chair_kKHb · 2023-10-27

**Recommendation:** Accept (Poster)
**Confidence:** 4

**Metareview:**

This work investigates a transition path sampling MCMC procedure in the latent space of a Normalizing-flow model, known as Boltzmann generator.  The authors propose a novel technique for transition path sampling (TPS) between two 3D states of molecular systems.

While the idea of latent space sampling within the context of BGs was already explored in the original paper from Noe et al., this study carries some additional insights that are of great relevance to the workshop.

Given the above and the common consensus of the referees, I recommend acceptance for this paper.